# Deregulated Gene Expression Profiles and Regulatory Networks in Adult and Pediatric RUNX1/RUNX1T1-Positive AML Patients

**DOI:** 10.3390/cancers15061795

**Published:** 2023-03-16

**Authors:** Peggy Kanellou, Ilias Georgakopoulos-Soares, Apostolos Zaravinos

**Affiliations:** 1Department of Hematology, Venizeleio General Hospital of Heraklion, 71409 Heraklion, Greece; 2Department of Biochemistry and Molecular Biology, Institute for Personalized Medicine, The Pennsylvania State University College of Medicine, Hershey, PA 17033, USA; 3Department of Life Sciences, School of Sciences, European University Cyprus, Nicosia 1516, Cyprus; 4Cancer Genetics, Genomics and Systems Biology Laboratory, Basic and Translational Cancer Research Center (BTCRC), Nicosia 1516, Cyprus

**Keywords:** acute myeloid leukemia, RUNX1/RUNX1T1 fusion

## Abstract

**Simple Summary:**

AML is a heterogeneous and complex disease. RUNX1/RUNX1T1 is a fusion oncogene resulting from the chromosomal translocation t(8;21) and plays a crucial role in AML. However, its impact on the transcriptomic profile of different age groups among AML patients is not completely understood. We aimed to investigate the deregulated gene expression profiles in RUNX1/RUNX1T1-positive AML patients, and compare their functions and regulatory networks between adult and pediatric patients. Our data corroborate that the RUNX1/RUNX1T1 fusion reprograms a large transcriptional network to establish and maintain leukemia via intricate protein–protein interactions and kinase-driven phosphorylation events.

**Abstract:**

Acute myeloid leukemia (AML) is a heterogeneous and complex disease concerning molecular aberrations and prognosis. RUNX1/RUNX1T1 is a fusion oncogene that results from the chromosomal translocation t(8;21) and plays a crucial role in AML. However, its impact on the transcriptomic profile of different age groups of AML patients is not completely understood. Here, we investigated the deregulated gene expression (DEG) profiles in adult and pediatric RUNX1/RUNX1T1-positive AML patients, and compared their functions and regulatory networks. We retrospectively analyzed gene expression data from two independent Gene Expression Omnibus (GEO) datasets (GSE37642 and GSE75461) and computed their differentially expressed genes and upstream regulators, using *limma*, *GEO2Enrichr*, and *X2K*. For validation purposes, we used the TCGA-LAML (adult) and TARGET-AML (pediatric) patient cohorts. We also analyzed the protein–protein interaction (PPI) networks, as well as those composed of transcription factors (TF), intermediate proteins, and kinases foreseen to regulate the top deregulated genes in each group. Gene ontology (GO) and Kyoto Encyclopedia of Genes and Genomes (KEGG) pathways enrichment analyses were further performed for the DEGs in each dataset. We found that the top upregulated genes in (both adult and pediatric) RUNX1/RUNX1T1-positive AML patients are enriched in extracellular matrix organization, the cell projection membrane, filopodium membrane, and supramolecular fiber. Our data corroborate that RUNX1/RUNX1T1 reprograms a large transcriptional network to establish and maintain leukemia via intricate PPI interactions and kinase-driven phosphorylation events.

## 1. Introduction

Acute myeloid leukemia (AML) is a heterogeneous disease in respect of molecular aberrations and prognosis. It is the second most common type of leukemia in adults and the most common type of acute leukemia. Nevertheless, AML is relatively rare and it accounts for approximately 1% of adult cancers in the United States, but nearly 2% of cancer-related deaths [1]. The Surveillance, Epidemiology, and End Results (SEER) Program reported annual incidence rates from 2010 and onward being consistently higher than 4.2 per 100,000 per year [2]. The overall AML incidence in childhood has also increased between 1975 and 2014 [3]. AML is defined by clonal expansion of malignant hematopoietic progenitor cells combined with differentiation arrest in the bone marrow [4]. Both epigenetic and genetic alterations have been implicated in the disease pathogenesis, with ~30–40% of the cases associated with recurrent chromosomal translocations, resulting in the generation of chimeric oncogenes [5].

In AML, drug resistance toward standard chemotherapeutic compounds is due to different factors, such as enhanced DNA repair, inhibition of cancer cell apoptosis, alteration of signaling pathways, alterations in drug metabolism enzymes, drug sequestration in intracellular organelles, and overexpression of efflux drug transporters [6,7]. In addition, overexpression of P-gp can lead to failure of the treatment or relapse. In fact, P-gp overexpression can result in highly aggressive AML clones, which need more intensive treatment and invasive procedures [6,7]. Multidrug resistance mechanisms also produce resistance to microtubule-targeting agents [8], such as vinca alkaloids and taxanes. Recently, three [1,2]oxazolo [5,4-e]isoindole derivatives showed antimitotic activity in HL-60R cells by inhibiting tubulin polymerization, suggesting that they could represent a valuable tool to overcome the MDR mechanism [9].

Runt-related transcription factor 1 (RUNX1) is crucial for the development of normal hematopoiesis [10], acting synergistically with ELF2 and ELF4 to transactivate the IL-3 and BLK promoters, respectively [11,12]. On the other hand, RUNX1 partner transcriptional co-repressor 1 (RUNX1T1) encodes a member of the myeloid translocation gene family that interact with DNA-bound transcription factors (TF) and recruit a variety of corepressors in order to achieve transcriptional repression. The t(8;21)(q22;q22) translocation is among the most frequent karyotypic aberrations in AML and produces the RUNX1/RUNX1T1 fusion oncogene. This chimeric protein is the most prevalent chromosomal rearrangement, especially in children and young adults [13]. However, ~50% of the patients relapse despite achieving complete remission [14,15]. The presence of t(8;21) establishes the diagnosis of AML, irrespective of blast count [16]. Interestingly, the clinical and prognostic features of t(8;21) differ between adults and children. On one hand, the t(8;21) is found in 1–7% of adults with newly diagnosed AML and the age at presentation is younger than that for the overall group of adults with AML [17]. In addition, AML with the t(8;21) has a favorable prognosis in adults. In pediatric AML, on the other hand, t(8;21) is the most frequent chromosomal abnormality. In contrast to adults, children with t(8;21) have poor outcomes, particularly when accompanied by additional cytogenetic abnormalities, such as del(9q), or gain of chromosome 4 [18]. The RUNX1/RUNX1T1 fusion product occupies >4000 genomic sites in hematopoietic cells and recruits several co-factors, with which it forms transcription–regulatory complexes via local chromatin remodeling and orchestrates cell differentiation, self-renewal, apoptosis and, ultimately, malignant transformation [19,20,21]. In addition, the RUNX1/RUNX1T1 oncogene was recently shown to regulate alternative RNA splicing and induce transcriptome re-organization in leukemic cells [22]. Nevertheless, despite the collective data on the molecular functions of the RUNX1/RUNX1T1 fusion product, its impact on the gene expression profile of leukemic cells is not completely understood.

To gain a better understanding of the regulatory functions of the RUNX1/RUNX1T1 fusion oncogene in the transcriptome of AML patients of different age groups, we investigated the upstream regulators of the differentially expressed genes (DEG) in two independent GEO datasets composed of adult and pediatric AML patients harboring the RUNX1/RUNX1T1 fusion. We validated our findings using two independent datasets, and then we annotated and extracted gene expression signatures to acquire a better insight into the link between the DEGs and leukemic cells that harbor the RUNX1/RUNX1T1 fusion, as well as to decipher differences between the two age groups. We focused on the deregulated TFs and the protein kinases that seem to act as “hubs” in the gene networks formed in RUNX1/RUNX1T1-positive AML cases.

## 2. Materials and Methods

### 2.1. Data Extraction and Gene Expression Analysis in RUNX1/RUNX1T1 AML Patients

We extracted gene expression data from two Gene Expression Omnibus (GEO) datasets. The first contained a total of 140 adult AML patients, 7 of which harbored the RUNX1/RUNX1T1 fusion (GEO accession, GSE37642; Platform, GPL570) [23,24,25,26]. The second contained 16 pediatric AML patients harboring the RUNX1/RUNX1T1 fusion out of a total of 48 AML patients (GSE75461; Platform, GPL17586) [27,28].

Differential gene expression between RUNX1/RUNX1T1-positive and -negative AML patients was identified using *limma* with cut-off |log_2_FC > 2| for upregulation and |log_2_FC < 1| for downregulation, along with an adjusted *p*-value of <0.05. We utilized the B-statistic to sort DEGs and run Gene Ontology (GO) enrichment analysis to functionally annotate the 50 most prevalent DEGs in RUNX1/RUNX1T1 AML [29,30]. The hypergeometric test was employed to uncover GO terms that were considerably enriched when compared to the entire human genome. We used the Fisher’s exact test to determine statistical significance and Benjamini–Hochberg (BH) correction for *p*-values, with a threshold of an adjusted *p*-value (adj-*p*) ≤ 0.05. We clustered together gene sets with similar GO terms, using Uniform Manifold Approximation and Projection (UMAP) [31] and highlighted the significantly enriched (adj. *p* < 0.05) GO terms for biological processes (GO-BP), molecular function (GO-MF), and cellular component (GO-CC). We calculated clusters with the Leiden algorithm [32] and created Figures using BokehJS 2.3.2.

The top deregulated genes and their identified upstream regulators were validated using two independent AML patient cohorts: the TARGET-AML project (https://ocg.cancer.gov/programs/target/projects/acute-myeloid-leukemia) [33] (accessed on 5 January 2023), containing a total of 2114 high-risk or hard-to-treat childhood cancers, and the Cancer Genome Atlas—Acute Myeloid Leukemia Study (TCGA-LAML) study, containing 200 clinically annotated adult cases of de novo AML [34]. The mRNA-seq data that we used (FPKM-UQ) were collected from an Illumina Hi-Seq 2000.

From the TARGET-AML study, we collected the following clinicopathological data: type of gene fusion, presence of the t(8;21) translocation, age at diagnosis (days), and treatment with gemtuzumab ozogamicin, as well as gene expression levels for *RUNX1*, *RUNX1T1*, *LINC00189*, *M1AP*, *ADCY7*, *TPPP3*, *ADARB1*, *TSPAN32*, *GPR114*, *PALM*, *SLCO5A1*, *HDX*, *SHANK1*, *SNORD9*, *EGLN1*, *HIPK2*, *MAPK1*, *CSNK2A1*, *MAPK3*, *MAPK14*, *SUZ12*, *PPARG*, *SRF*, *CBX3*, *SALL4*, *CF3*, *AR*, *HNF4A*, *NFE2L2*, *EZH2*, *TP63*, *ESR1*, *SMC3*, *STAT3*, *KLF4*, *GATA1*, *CTCF*, *PPARD*, *TP53*, *CDK1*, *GSK3B*, *CDK2*, *AKT1*, *CDK4*, and *MAPK8*.

From the TCGA-LAML study, we collected the following clinicopathological data: cytogenetic abnormality, prior treatment diagnoses, age at diagnosis, as well as gene expression levels for *RUNX1T1*, *POU4F1*, *CACNA2D2*, *FBLN5*, *CAV1*, *CLEC2L*, *SIPA1L2*, *GPM6B*, *CD19*, *IL5RA*, *PALM*, *ITGB4*, *FAM81A*, *ROBO1*, *FOXL1*, *TRH*, *IGBP1*, *RNF39*, *FRMD1*, *INPPL1*, *FSCN2*, *NUBPL*, *UQCRQ*, *C12orf10*, *LINC00115*, *LINC00189*, *M1AP*, *ADCY7*, *TPPP3*, *ADARB1*, *TSPAN32*, *GPR114*, *SLCO5A1*, *HDXS*, *HANK1*, *SNORD9*, *EGLN1*, *CSNK2A1*, *AKT1*, *MAPK3*, *STAT3*, *SMAD4*, *IRF1*, *TCF3*, *SOX2*, *EZH2*, *PPARD*, *FOSL2*, *ATF2*, *ELF1*, *E2F1*, *ERG*, *MYOD1*, *FOXP2*, *TCF7L2*, *GABPA*, *FOXA1*, *CDK1*, *MAPK14*, *MAPK8*, *MAPK1*, and *SUZ12*.

The TARGET-AML study contains 284 AML patients with the RUNX1-RUNX1T1 fusion, 46 of whom received treatment with gentuzumab ozogamicin (the other 122 did not). On the other hand, the TCGA-AML cohort does not contain a t(8;21), but two patients with 21,9(9;22), six patients with t(15;17), six patients with t(4;11), eight patients with t(9;22), and thirty-eight AML patients with genetic abnormality in chr21.

### 2.2. Upstream Regulators of the Deregulated Genes in RUNX1/RUNX1T1 AML Patients

We used *Expression2Kinases* (X2K) to infer the upstream regulatory networks of the DEGs in RUNX1/RUNX1T1 AML. We further performed transcription factor (TF) enrichment analysis, expansion analysis of the protein–protein interaction (PPI) networks, kinase enrichment, and we inferred the gene networks governing the expression of the DEGs in RUNX1/RUNX1T1 AML patients [35].

### 2.3. Statistical Analysis

We assessed statistical differences in gene expression between RUNX1/RUNX1T1-positive and RUNX1/RUNX1T1-negative AML samples through the Mann–Whitney U test, with an adj. *p* < 0.05 as a threshold for statistical significance. The expression levels of the genes that were significantly deregulated in both validation cohorts (TCGA-AML and TARGET-AML) were compared using the Mann–Whitney U test with Bonferroni correction (threshold, *p* ≤ 0.05).

## 3. Results

### Deregulated Genes and Functional Analysis in RUNX1/RUNX1T1 AML Patients

We first detected the significantly deregulated genes in adult RUNX1/RUNX1T1 AML patients (GSE37642), and focused on the top (and bottom) 50 DEGs (Figure 1). As expected, *RUNX1T1*, *POU4F1*, *CACNA2D2*, *FBLN5,* and *CAV1* were among the top upregulated genes in RUNX1/RUNX1T1 AML patients, followed by *CLEC2L*, *SIPA1L2*, *GPM6B*, *CD19*, *IL5RA*, *PALM*, *ITGB4*, *FAM81A*, *SLCO5A1*, *ROBO1*, *FOXL1,* and *TRH* (Appendix A and Figure 2). The top 50 upregulated genes were primarily enriched in *axonogenesis*, *axon guidance*, *extracellular matrix organization*, *negative regulation of cell differentiation, and extracellular matrix assembly* (GO-BP). They were also overrepresented in *epidermal growth factor receptor binding*, *transmembrane receptor protein tyrosine kinase activity*, *neuregulin binding*, *neurotrophin TRK receptor binding, and neurotrophin TRKA receptor binding* (GO-MF), as well as in *collagen-containing extracellular matrix*, *cell projection membrane*, *sarcolemma*, *basement membrane, and membrane raft* (Appendix A).

In contrast, the bottom 50 DEGs included IGBP1, RNF39, FRMD1, INPPL1, FSCN2, NUBPL, UQCRQ, C12orf10, LINC00115, and LOC100129198, and were enriched in mitochondrial electron transport, ubiquinol to cytochrome c, aerobic electron transport chain, mitochondrial ATP synthesis coupled electron transport, mitochondrial respiratory chain complex I assembly, and NADH dehydrogenase complex assembly (GO-BP). They were also enriched in syntaxin binding, fucose binding, NAADP-sensitive calcium-release channel activity, immunoglobulin receptor binding, and ubiquinol-cytochrome-c reductase activity (GO-MF), as well as in mitochondrial respiratory chain complex III, pseudopodium, and actin cytoskeleton (GO-CC) (Appendix A). Of note, UMAP did not discriminate RUNX1/RUNX1T1-positive from RUNX1/RUNX1T1-negative adult AML patients.

We similarly detected the significantly deregulated genes in pediatric RUNX1/RUNX1T1 AML patients (Figure 3 and Appendix A). Among the top upregulated genes, we identified *LINC00189*, *RUNX1T1*, *M1AP*, *ADCY7*, *TPPP3*, *ADARB1*, *TSPAN32*, *GPR114*, *PALM*, *SLCO5A1,* and *HDX*; whereas, among the top downregulated genes, we highlighted *SHANK1*, *linc-RXFP2-1*, *SNORD9,* and *EGLN1*.

The first were primarily enriched in extracellular matrix (ECM) organization, protein polymerization, and cellular protein catabolic process (GO-BP). They were also enriched in choline transmembrane transporter activity, double-stranded RNA adenosine deaminase activity, and diphosphotransferase activity (GO-MF), as well as in an intrinsic component of the cytoplasmic side of the plasma membrane, the cell projection membrane, an integral component of the lysosomal membrane and microfibril (GO-CC) (Appendix A).

Interestingly, the GO-terms *ECM organization* (GO-BP), *cell projection membrane; filopodium membrane, and supramolecular fiber* (GO-CC) were commonly found among the top 10 GO terms in both age groups of RUNX1/RUNX1T1-positive AML.

In contrast, the bottom 50 DEGs were highly enriched in membrane depolarization during cardiac muscle cell action potential, regulation of ventricular cardiac muscle cell membrane repolarization, cardiac muscle cell contraction, and regulation of cardiac muscle cell membrane repolarization (GO-BP). Furthermore, they were enriched in phosphatidylinositol 3-kinase regulatory subunit binding, voltage-gated calcium (and sodium) channel activity involved in cardiac muscle cell action potential, ankyrin repeat binding, and phosphatidic acid transfer activity (GO-MF), as well as in sperm flagellum and 9 + 2 motile cilium (GO-CC) (Appendix A and Figure 4).

In order to gain a better insight into the critical hubs that orchestrate the expression of the DEGs, we constructed the PPI networks. In the adult AML patient cohort (GSE37642), we found that SUZ12, SOX2, EZH2, SMAD4, and STAT3 (among other TFs), as well as MAPK14, CSNK2A1, MAPK1, CDK1, AKT1, and MAPK3 (among other kinases) regulate the co-upregulated genes upstream. Of these, we noticed that SUZ12, STAT3, SMAD4, IRF1, TCF3, SOX2, EZH2, and PPARD act as hubs in the PPI expansion network (Figure 5a).

On the other hand, the co-downregulated genes seem to be orchestrated by FOSL2, ATF2, ELF1, E2F1, SUZ12, ERG, MYOD1, FOXP2, TCF7L2, GABPA, FOXA1, and other TFs, as well as by the kinases CDK1, MAPK14, MAPK8, JNK1, and MAPK1. Of these, E2F1, ATF2, MYID1, TCF7L2, ERG, and SUZ12 appear to act as hub proteins in the PPI network (Figure 5b).

Similarly, we calculated the responsible upstream regulators for the co-deregulated genes among pediatric AML patients (GSE75461). We found RUNX1, RCONR1, NFE2L2, SUZ12, PPARG, SRF, CBX3, SALL4, TCF3, among other TFs, but also the kinases ERK1/2, HIPK2, CK2APLHA, MAPK1, CSNK2A1, and MAPK3/14, among others. Of these, SRF, PPARG, RUNX1, NFE2L2, TCF3, CBX3, IRF8, SUZ12, and RCOR1 were found to act as hub TFs in the PPI expansion network (Figure 6a).

Additionally, among the co-downregulated genes, we found AR, HNF4A, NFE2L2, EZH2, TP63, ESR1, SMC3, STAT3, KLF4, SXO2, GATA1, CTCF, PPARD, and TP53, among other TFs, and the kinases CSNK2A1, MAPK14, CDK1, MAPK1/3, HIPK2, GSK3B, CDK2, ERK1/2, DNAPK, AKT1, CDK4, GSK3BETA, and MAPK8. Of these, we detected EZH2, NFE2L2, HNF4A, STAT3, TP63, and KLF4 as hub proteins in the PPI network (Figure 6b).

We also validated the top deregulated genes and their upstream regulators in two independent AML patient cohorts: the TCGA-LAML and TARGET-AML studies, containing adult and pediatric AML patients, respectively (Figure 7 and Appendix A). In specific, in the TCGA-LAML dataset, we found significant upregulation for *TRH*, *POU4F1*, *TCF3*, *CAV1*, *PALM*, *ROBO1*, *SIPA1L2*, *SHANK1*, *IL5RA*, *CD19*, *CACNA2D2*, *ITGB4*, *SLCO5A1*, *M1AP*, *FBLN5*, *LINC00189,* and *EGL1* in RUNX1/RUNX1T1-positive patients; whereas, *ADCY7*, *TSPAN32*, *ADAB1*, *PPARD*, and *FOXL1* were significantly downregulated in them compared to RUNX1/RUNX1T1 = negative patients (Figure 7a). In the TARGET-AML dataset, on the other hand, we found significant upregulation for *TCF3*, *RUNX1T1*, *PALM*, *SHANK1*, *CSNK2A1*, *TPPP3*, *SLCO5A1*, *M1AP*, and *LINC00189* in RUNX1/RUNX1T1-positive patients; whereas, *KLF4*, *ADCY7*, *TSPAN32*, *ADARB1*, *PPARD*, *HDX*, and *ESR1* were significantly downregulated in them compared to RUNX1/RUNX1T1 = negative patients (Figure 7b). Considering the commonly deregulated genes between the two AML datasets, we found that *RUNX1*, *INPPL1*, *MAPK1*, *SMC3*, *TP53*, *ERG*, *CDK4*, *ADCY7*, *CDK1*/*2*, and *ADARB1*, among others, were significantly upregulated in the adult AML cohort (TCGA-LAML); whereas *ELF1*, *FOSL2*, *IGBP1*, *STAT3*, *SRF*, *EZH2*, *NFE2L2*, *ATF2*, *SUZ12*, *IRF1*, *CTCF*, *UQCRQ*, *GSK3B*, *SNORD9*, *PPARD*, *CD19*, and *GPM6B,* among others, were upregulated in the pediatric AML cohort (TARGET-AML) (Figure 7c).

## 4. Discussion

Leukemia is characterized by somatic mutations and genetic rearrangements that impede signal transduction and expression programs, resulting in the disruption of chromatin modifiers and transcription factors. In AML, transcriptional and epigenetic reprogramming can be seen in the RUNX1 transcription factor and its oncogenic derivative RUNX1/RUNX1T1 [36]. The RUNX1/RUNX1T1 fusion has been shown to lead to aberrations in various intracellular pathways, including the downregulation of tumor suppressors or the activation of oncogenes [37]. Nevertheless, the role of the RUNX1/RUNX1T1 fusion oncogene is not completely understood in the AML transcriptome of different age groups of patients. Here, we explored the gene networks governing the top deregulated genes in adult and pediatric RUNX1/RUNX1T1 AML patients.

RUNX1 is a TF responsible for the differentiation of hematopoietic cells and the development of myeloid cells [36,38]. RUNX1T1 is a transcriptional co-repressor [39,40]; however, its role is less well understood [41]. The t(8;21)(q22;q22) translocation puts these two side by side, with chromosomal breakpoints clustering within intron 5 of *RUNX1* and intron 1 of *RUNX1T1* [42,43]. The expression of the RUNX1/RUNX1T1 fusion protein alone is not sufficient to induce leukemia, but it is a complex process which includes the deregulation of various pathways [37]. The overexpressed RUNX1T1 does not interact directly with DNA, but it is recruited by the TFs GFI1 and BCL6. In turn, RUNX1T1 recruits several corepressors, including histone deacetylases (HDACs), SMRT, N-CoR, and mSin3, with which it mediates the transcriptional repression of RUNX1-target hematopoietic genes [37]. In turn, this leads to an arrest of cell differentiation, an increase in cell survival, and the promotion of leukemogenesis [44,45]. The RUNX1/RUNX1T1 fusion also blocks the hematopoietic TFs PU.1, GATA1, and CEBPA, which regulate myeloid differentiation [46,47,48]. It further inhibits the tumor suppressors RUNX3 and NF1 [49,50], but also HIF1a and MEIS2 [51], all of which are overexpressed in AML t(8;21) patients. The RUNX1/RUNX1T1 fusion protein also inhibits apoptosis, since the antiapoptotic proteins BCL-2 and BCL-XL are often upregulated in RUNX1/RUNX1T1 AML patients [52,53]. It also impairs the activity of genes taking part in DNA repair (e.g., POLE and OGG1) [54,55].

Here, we explored the upstream regulators of the DEGs in adult and pediatric AML patients with the RUNX1/RUNX1T1 fusion oncogene. We found that apart from *RUNX1T1*, which is known to be highly expressed in RUNX1/RUNX1T1 AML, *POU4F1*, *CACNA2D2*, *FBLN5,* and *CAV1* are also upregulated in adult RUNX1/RUNX1T1 AML patients, followed by *CLEC2L*, *SIPA1L2*, *GPM6B*, *CD19*, *IL5RA*, *PALM*, *ITGB4*, *FAM81A*, *SLCO5A1*, *ROBO1*, *FOXL1,* and *TRH*. 

SLCO5A1 is a member of the solute carrier organic anion transporters [56]. Organic anion transporting polypeptides play a key role in the uptake and distribution of endogenous compounds and xenobiotics [57], and, in concordance with our findings, they are deregulated in several types of cancer, suggesting that they play a potential pathogenic role during cancer development and progression. [58]. Furthermore, the TF POU4F1 is known to be overexpressed in RUNX1/RUNX1T1 leukemia [59]. CAV1 is the main coat protein of caveolae and interacts with various cellular proteins in order to regulate cell-signaling. It functions as a tumor suppressor in some cancers, but in others, it promotes cell survival, adhesion, and migration [60,61]. In leukemia, CAV1 seems to be involved in the modulation of VEGF-induced redox signal transduction [62].

In contrast, in pediatric RUNX1/RUNX1T1 AML, the top upregulated genes were *LINC00189*, *RUNX1T1*, *M1AP*, *ADCY7*, *TPPP3*, *ADARB1*, *TSPAN32*, *GPR114*, *PALM*, *SLCO5A1,* and *HDX*. Of these, the membrane protein ADCY7 has been shown to support leukemogenesis by decreasing apoptosis [63]. Adenosine deaminase RNA-specific B1 (ADARB1) is an RNA-editing enzyme involved in the development of various cancer types, including AML [64,65]. Particularly, ADAR1 plays an important role in adult hematopoiesis via its RNA editing activity in hematopoietic progenitor cells [66].

Several G protein-coupled receptor members (GPCRs) are also overexpressed and enriched in adhesion receptor subfamilies (e.g., GPR114 [67]); whereas, others are downregulated in AML. It is also worth noting that among the upregulated genes that we found, it is the first time, to our knowledge, that *FBLN5*, *CACNA2D2*, *M1AP*, *TPPP3,* and *TSPAN32* are mentioned to be upregulated in AML, indicating that perhaps other pathways are also likely implicated in the disease.

Of these, FBLN5 is an anti-oncogene that belongs to the family of fibulins, and its aberrant expression has been reported in lung, breast, hepatocellular, and other types of cancer [68,69,70,71,72]. Moreover, expression deficiency of *CACNA2D2*, which encodes for a new subunit of the Ca^2+^-channel complex, has been postulated as a possible link in the pathogenesis of lung cancer [73], similar to *M1AP* [74]. Furthermore, TPPP3 mediates the dynamics and stability of microtubules and is associated with multiple immune-related pathways in various types of cancer [75]. In addition, TSPAN32 is a member of the tetraspanins, which contribute to tumor growth through angiogenesis, immunological function, platelet coagulation, and infection [76,77]. On the other hand, *LINC00189* and *PALM* are rarely reported in human diseases.

Interestingly, we found three genes being commonly upregulated between pediatric and adult RUNX1/RUNX1T1 AML patients. These are *RUNX1T1*, *PALM,* and *SLCO5A1*, and play a role in the regulation of the dopamine receptor signaling pathway, the negative regulation of adenylate cyclase activity, and the sodium-independent organic anion transport, among others.

We also identified the top 50 downregulated genes in adult RUNX1/RUNX1T1 AML patients. These included *IGBP1*, *RNF39*, *FRMD1*, *INPPL1*, *FSCN2*, *NUBPL*, *UQCRQ*, *C12orf10*, *LINC00115,* and *LOC100129198*; whereas, in pediatric RUNX1/RUNX1T1 AML patients, they included *SHANK1*, *linc-RXFP2-1*, *SNORD9,* and *EGLN1*.

Similar to the CBFB/MYH11 fusion, the RUNX1/RUNX1T1 fusion protein is associated with unique TF networks, which have some common nodes across all AML types and are post-translationally modified [43,78]. In the present study, we constructed the PPI networks containing the critical hub proteins. Of the upstream regulators for the co-upregulated genes in the adult AML patient cohort (GSE37642), we found that SUZ12, STAT3, SMAD4, IRF1, TCF3, SOX2, EZH2, and PPARD act as hubs. On the other hand, the co-downregulated genes were significantly associated with the hub TFs, being E2F1, ATF2, MYID1, TCF7L2, ERG, and SUZ12.

In addition, we calculated the responsible upstream regulators of the co-deregulated genes between the two datasets. Our findings show that SRF, PPARG, RUNX1, NFE2L2, TCF3, CBX3, IRF8, SUZ12, and RCOR1 seem to act as hub TFs in the formed PPI network of the upregulated genes; whereas, EZH2, NFE2L2, HNF4A, STAT3, TP63, and KLF4 orchestrate the co-downregulated genes. The co-expression networks that our data highlight corroborate that the RUNX1/RUNX1T1 fusion protein reprograms a large transcriptional network.

Interestingly, TCF3 was a common upstream regulator in both cohorts. Furthermore, STAT3 and EZH2 were found to be upregulated in the adult AML cohort, but downregulated in the pediatric one, suggesting that they act differently between pediatric adult AML patients.

The expression pattern of TFs seems to have an important impact on the PPI network. The polycomb group of proteins (PcG) is a major class of epigenetic regulators that repress target genes involved in cell cycle regulation and differentiation [79,80]. PcGs can act either as tumor-suppressors or oncogenes, depending on the cell context or their interaction partners, and their abnormal expression or mutations have been shown to occur regularly across various blood cancers [81].

A large number of studies have recently revealed that AML patients have repeated mutations in several important epigenetic mediators [82]. Mutations in genes involved in DNA methylation, such as DNA methyltransferase 3A (DNMT3A) and isocitrate dehydrogenase 1 and 2 (IDH1/2), are the most common. The majority of these mutations affect the hematopoietic stem cells (HSC) compartment and alter hematopoietic differentiation, in some cases leading to myelodysplasia, stem cell expansion, or other preleukemic conditions [83,84,85]. In addition, mutations in other epigenetic modifiers, such as additional sex combs-like (ASXL1) and enhancer of zeste homolog 2 (EZH2), contribute to the occurrence of AML. Nevertheless, epigenetic mutations alone are insufficient to transform HSCs, implying that mutations must be acquired sequentially [25]. In Npm1c/Dnmt3a mutant knock-in mice, a model of AML development, the use of a small molecule (VTP-50469) could indeed reverse myeloid progenitor cell self-renewal prior to leukemia transformation, suggesting that individuals at high risk of developing AML may benefit from preventive targeted epigenetic therapy [86].

The effect of RUNX1 on target gene expression is highly context-dependent, determined by the composition of the transcriptional complexes in which RUNX1 functions at a particular gene. These complexes contain transcriptional cofactors as well as epigenetic modifiers, which are recruited to gene promoters as well as other regulatory regions, and by doing so, RUNX1 can influence both transcriptional activity as well as the epigenetic status of target genes [36].

RUNX1 has been found to complex with an array of epigenetic modifiers, with the outcome for both RUNX1 function and target gene activity dependent on the balance of activating and repressive factors associated with RUNX1 at a particular time. These studies thus demonstrate an idea of RUNX1 itself as a target of epigenetic modifiers, which post-translationally modify the RUNX1 protein, altering its coactivator/corepressor interactions, thus influencing its transcriptional activity. In addition, RUNX1 recruits these coactivators and corepressors to target genes, resulting in the modification of the chromatin environment, thus affecting gene activity at this second level. An outstanding example of this multi-leveled interaction between RUNX1, epigenetic modifiers, and the epigenome is demonstrated in the recently described interaction of RUNX1 with protein arginine methyltransferase 6 (PRMT6) [36].

What is more, our findings regarding the deregulated expression of *EZH2* in AML agree with previous reports, suggesting that its overexpression (apart from mutation) promotes cell proliferation in hematologic and other malignancies. EZH2 controls the expression of genes involved in stem cell maintenance and differentiation and its down-regulation inhibits apoptosis, affects the expression of MAD2 and CDC20, and promotes chromosomal instability in AML cells [87,88,89,90,91,92]. Our findings also highlight the deregulated expression of SUZ12, another component of PRC2, responsible for the transcriptional repression of genes via methylation [93,94]. The complete loss of SUZ12 leads to failed hematopoiesis, whereas partial loss of PRC2 enhances the self-renewal of HSCs [79].

Furthermore, we shed light on the Signal Transducer and Activator of Transcription 3 (STAT3), another TF that is crucial in leukemogenesis as it blocks the differentiation of myeloid cells [95,96]. STAT3 upregulation is associated with protection from apoptosis [97]. In concordance, AML cells expressing the *RUNX1*/*RUNX1T1* fusion have been shown to be sensitive to the inbitition of the JAK-STAT pathway [48]. Of note, STAT3 was upregulated in the adult AML cohoort and downregulated in the pediatric AML cohort. This discrepancy can perhaps be explained by the fact that STAT3 has heterogeneous functions and although it is frequently described as an oncogene, its opposite role as a tumor expressor most likely depends on the expression of its various gene isoforms [98,99,100,101,102,103].

## 5. Conclusions

To conclude, our data corroborate that hematopoiesis is strictly regulated by different levels of intercellular signaling, and is affected by various intricate transcriptional regulations. The RUNX1/RUNX1T1 fusion aberrates many intracellular pathways that involve tumor suppressors and oncogenes. The regulatory networks governing the top deregulated genes in adult and pediatric RUNX1/RUNX1T1 AML patients include several transcription factors, intermediate proteins and kinases, which orchestrate the establishment and maintenance of leukemia. The most upregulated genes in adult RUNX1/RUNX1T1-positive AML were *RUNX1T1***,**
*POU4F1***,**
*CACNA2D2***,**
*FBLN5,* and *CAV1*, which play a role in axonogenesis, ECM organization, and negative regulation of cell differentiation and ECM assembly. The top downregulated genes in this group included *IGBP1***,**
*RNF39***,**
*FRMD1***,**
*INPPL1***,**
*FSCN2***,**
*NUBPL,* and *UQCRQ*, among others, playing a role in mitochondrial electron transport, ubiquinol to cytochrome c, aerobic electron transport chain, and mitochondrial ATP synthesis coupled electron transport. The most upregulated genes in pediatric RUNX1/RUNX1T1-positive AML included *RUNX1T1***,**
*LINC00189***,**
*M1AP***,**
*ADCY7***,**
*TPPP3***,**
*ADARB1,* and *TSPAN32*, playing key role in ECM organization, protein polymerization, and cellular protein catabolic processes. Finally, the top downregulated genes in this group involved *SHANK1*, linc-*RXFP2-1*, *SNORD9,* and *EGLN1*, being primarily enriched in membrane depolarization during cardiac muscle cell action potential.

## Figures and Tables

**Figure 1 cancers-15-01795-f001:**
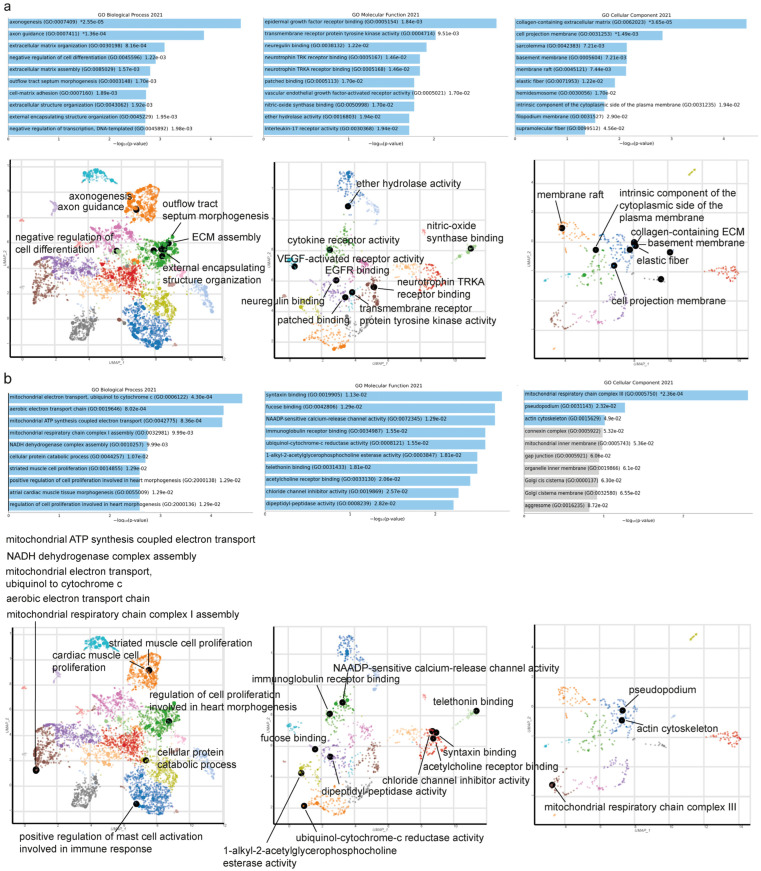
Bar charts (upper panel) depict the top 10 enriched GO terms in the upper (**a**) and lower (**b**) 50 deregulated genes in adult RUNX1/RUNX1T1 AML patients (GSE37642 dataset), along with their corresponding *p*-values (GO Biological Process; GO Molecular Function; GO Cellular Component). Statistically significant GO terms (adj. *p* < 0.05) are depicted with an asterisk (*). Scatterplots below show clusters of similar gene sets, the significantly enriched terms of which are denoted.

**Figure 2 cancers-15-01795-f002:**
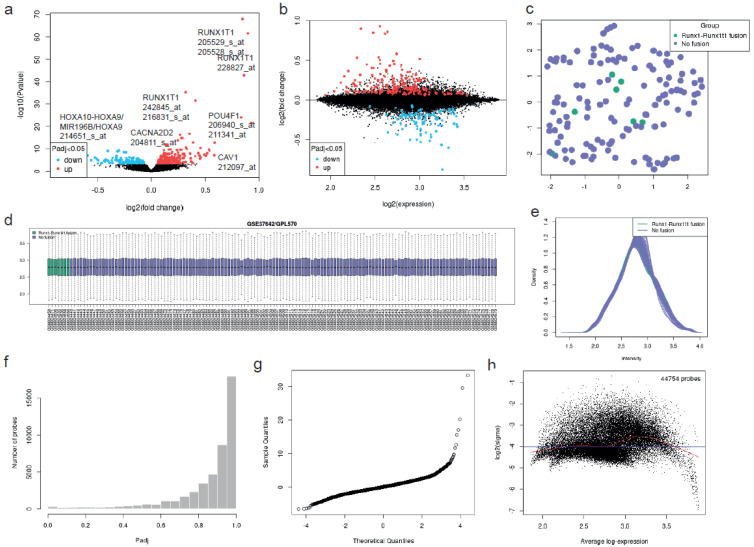
GSE37642 dataset analysis between RUNX1/RUNX1T1 fusion and RUNX1 AML patients. (**a**) The volcano plot depicts the statistical significance (−log_10_ *p*-value) against the magnitude of change (log_2_ fold change) across the differentially expressed genes in adult RUNX1/RUNX1T1 AML patients. Highlighted genes are significantly deregulated (adj. *p*-value ≤ 0.05) (red = upregulated, blue = downregulated). (**b**) The mean difference (MD) plot displays the log_2_ fold change versus the average log_2_ expression of the differentially expressed genes in adult RUNX1/RUNX1T1 AML patients. (**c**) UMAP clustering shows that adult AML samples are related to each other irrespective of the RUNX1/RUNX1T1 fusion. (**d**) Distribution of the log transformed and normalized gene expression values and density curves (**e**) across all adult AML samples. The median-centered values indicate that the data are normalized and cross-comparable. (**f**) The histogram depicts the distribution of the *p*-values. (**g**) Moderated t-statistic quantile-quantile (q-q) plot. (**h**) Mean-variance trend plot, in which each dot represents a gene, and it shows the mean–variance relationship of the genes in the dataset. The blue line is constant variance approximation.

**Figure 3 cancers-15-01795-f003:**
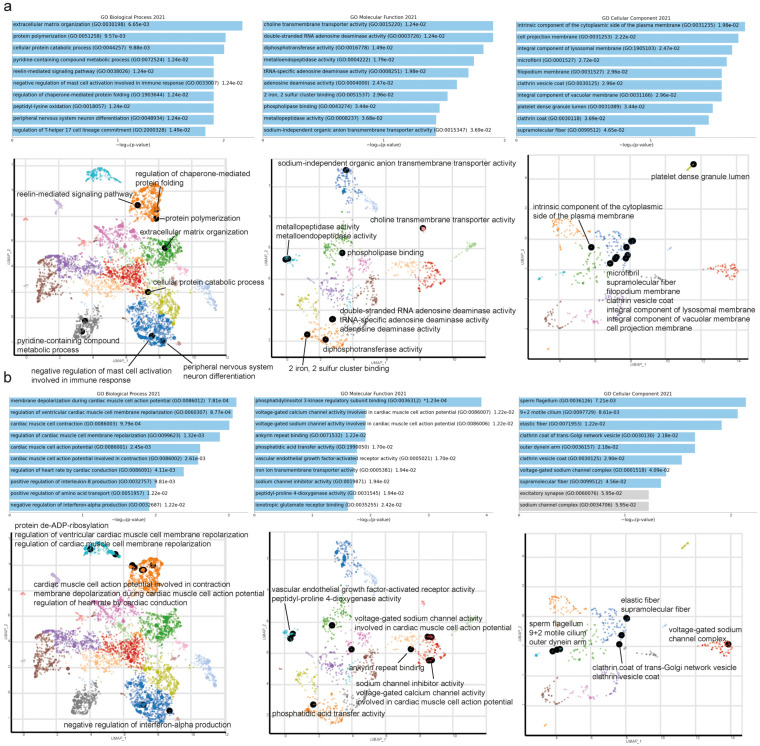
Bar charts (upper panel) show the GO terms being significantly enriched in the upper (**a**) and lower (**b**) 50 deregulated genes in pediatric RUNX1/RUNX1T1 AML patients (GSE75461 dataset), along with their corresponding *p*-values. Statistically significant GO terms (adj. *p* < 0.05) are depicted with an asterisk (*). Scatterplots below show clusters of similar gene sets, the significantly enriched terms of which are denoted.

**Figure 4 cancers-15-01795-f004:**
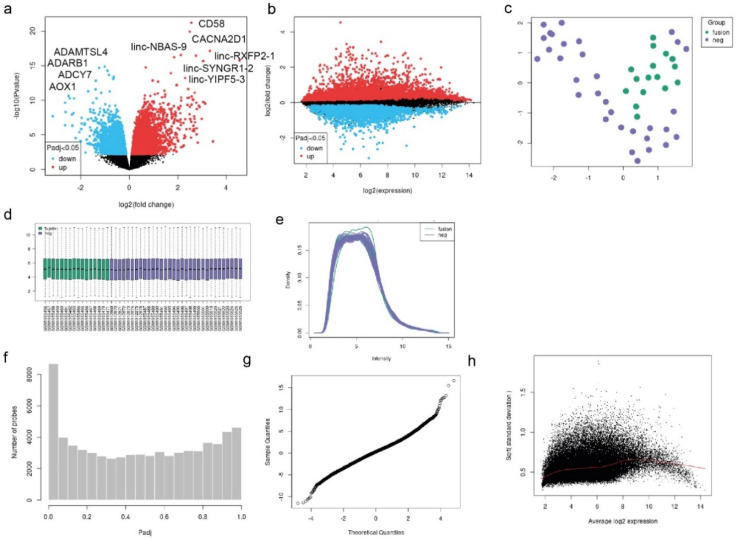
GSE75461 dataset analysis between RUNX1/RUNX1T1 fusion and RUNX1 AML patients. (**a**) The volcano plot depicts the statistical significance (−log_10_ *p*-value) against the magnitude of change (log_2_ fold change) across the differentially expressed genes in pediatric RUNX1/RUNX1T1 AML patients. Highlighted genes are significantly deregulated (adj. *p*-value ≤ 0.05) (red = upregulated, blue = downregulated). (**b**) The mean difference (MD) plot displays the log_2_ fold change versus average the log_2_ expression of the differentially expressed genes in pediatric RUNX1/RUNX1T1 AML patients. (**c**) UMAP clustering discriminated RUNX1/RUNX1T1-positive from RUNX1/RUNX1T1-negative pediatric AML samples. (**d**) Distribution of the log transformed and normalized gene expression values and density curves (**e**) across all pediatric AML samples. The median-centered values indicate that the data are normalized and cross-comparable. (**f**) The histogram depicts the distribution of the *p*-values. (**g**) Moderated t-statistic quantile-quantile (q-q) plot. (**h**) Mean-variance trend plot, as explained above.

**Figure 5 cancers-15-01795-f005:**
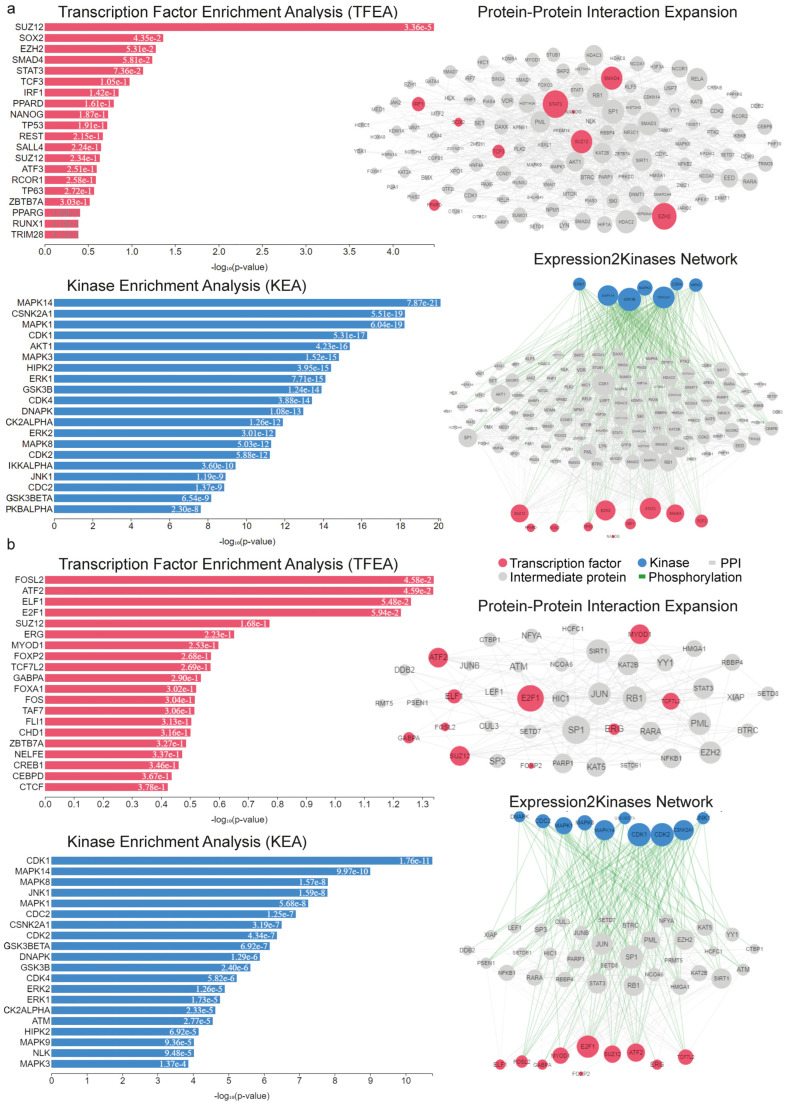
Networks of upstream regulators for most up- (**a**) or downregulated (**b**) genes in adult RUNX1/RUNX1T1 AML patients (GSE37642 dataset).

**Figure 6 cancers-15-01795-f006:**
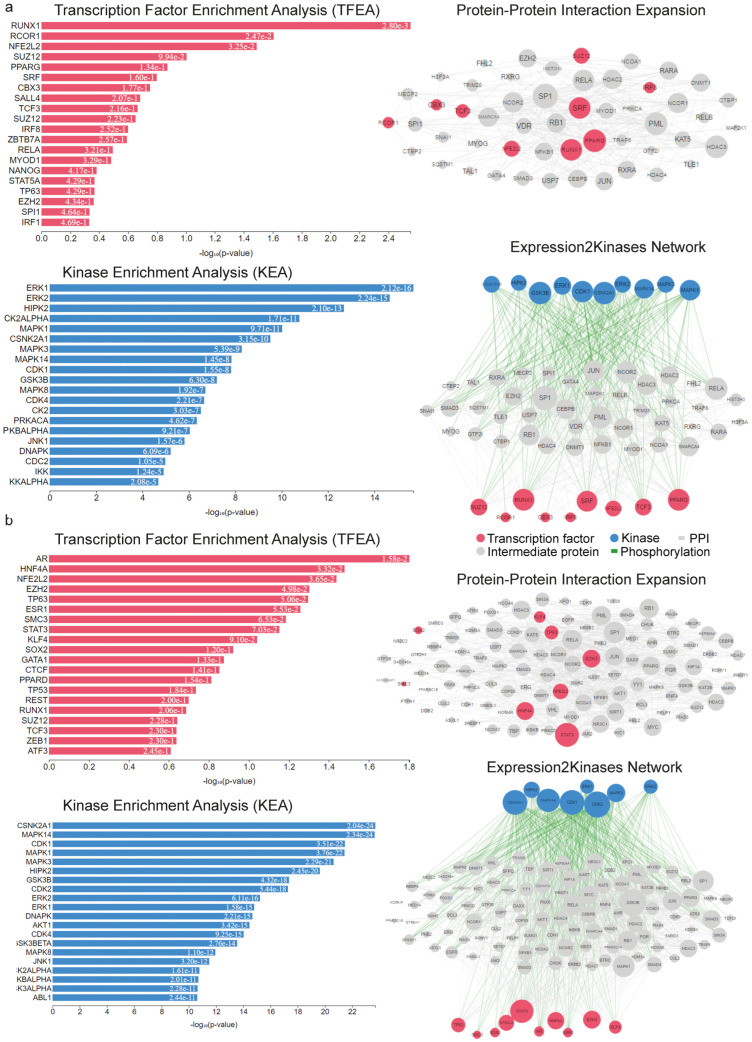
Upstream regulatory networks for the up- (**a**) and downregulated (**b**) genes in pediatric RUNX1/RUNX1T1 AML patients (GSE75461 dataset).

**Figure 7 cancers-15-01795-f007:**
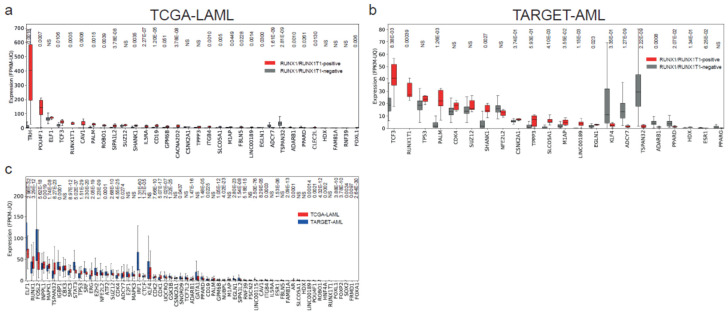
Expression levels of the top deregulated genes and their upstream regulators in RUNX1/RUNX1T1-positive and RUNX1/RUNX1T1-negative adult (**a**) or pediatric (**b**) AML patients. Data were extracted from the TCGA-LAML (adult AML) and TARGET-AML (pediatric AML) projects, respectively. Comparison of the expression levels between the top deregulated genes and their upstream regulators, between pediatric (TARGET-AML) and adult (TCGA-LAML) AML patients (**c**). The BH-adjusted *p*-values of all statistically significant differences between different groups are depicted on top of each subfigure. NS, not-significant *p*-value (*p* > 0.05).

## Data Availability

All data generated or analyzed during this study are included in this published article [and its Appendix A].

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
