# Peer review of "Deregulated Gene Expression Profiles and Regulatory Networks in Adult and Pediatric RUNX1/RUNX1T1-Positive AML Patients"

_cancers, 2023, doi:10.3390/cancers15061795_

Round 1

Reviewer 1 Report (Previous Reviewer 1)

The authors have accessed two large public datasets of gene expression data for adult and pediatric AML. Rather than applying their gene finding strategies to the large datasets, which would provide more statistical power, they seem to have used them for validation of the genes they found with the very small data sets. Furthermore they did not appear to modify their conclusions based on the new validation data, since there are genes described in the discussion as being upregulated that are shown to be downregulated in the new data. Examples include ADCY7 and ADARB1 (line 340 and figure 7b). Lastly, the purpose for comparing gene expression between those who received gemtuzumab and those who did not is not provided. Since the cells being studied were collected from patients before any treatment was given, this does not seem to be a meaningful comparison.

Author Response

Reviewer #1

The authors have accessed two large public datasets of gene expression data for adult and pediatric AML. Rather than applying their gene finding strategies to the large datasets, which would provide more statistical power, they seem to have used them for validation of the genes they found with the very small data sets. Furthermore they did not appear to modify their conclusions based on the new validation data, since there are genes described in the discussion as being upregulated that are shown to be downregulated in the new data. Examples include ADCY7 and ADARB1 (line 340 and figure 7b). Lastly, the purpose for comparing gene expression between those who received gemtuzumab and those who did not is not provided. Since the cells being studied were collected from patients before any treatment was given, this does not seem to be a meaningful comparison.

Author response: We agree with the reviewer’s comment that there should have been a reverse strategy followed from the beginning; i.e. an initial study on the largest datasets (TARGET-AML and TCGA-LAML) followed by a validation study on the GEO datasets. However, we have amended our work gradually during the peer-review process, as per the suggestions made by all three reviewers. Reviewer’s #1 suggestion to include the TARGET-AML and TCGA-LAML studies was satisfied in the second round of peer reviewing, and we are afraid that a reverse in the order of our analysis at this point, would demand a complete re-writing of our manuscript. Furthermore, it is true that given the higher statistical power of the two largest datasets (TARGET-AML and TCGA-LAML), one would expect to find discrepancies in the gene expression profiles of RUNX1/RUNX1T1-positive AML patients, across all datasets (TCGA/TARGET and GEO studies); such as in the case of ADCY7 and ADARB1. Therefore, we have now diminished the importance of these two specific findings, and we have excluded them from the Discussion, as proposed by the reviewer. In addition, we have now modified our conclusions based on the new validation data, as per the reviewer’s suggestion. Finally, we have excluded the comparison of gene expression between patients who received gemtuzumab and those who did not, as per the reviewer’s suggestion. We have also deleted the corresponding subfigure 7d.

Reviewer 2 Report (Previous Reviewer 3)

The authors revised the manuscript according to my comments and the paper can be accepted for publication.

Author Response

We would like to thank the reviewer for his/her insightful comments, which helped us ameliorate our manuscript. We are grateful for the endorsement of our work.

This manuscript is a resubmission of an earlier submission. The following is a list of the peer review reports and author responses from that submission.

Round 1

Reviewer 1 Report

This is a data mining report. The authors accessed publicly available gene expression data for adults and children with t(8;21) AML. They identified DEGs between t(8;21)+ and negative independently for the two age groups, and then used various programs to infer regulatory networks, protein-protein interactions, etc. The datasets used are rather small. In the adult dataset there were only 7 patients with the fusion of interest, and only 16 in the pediatric data. Despite all the analysis reported, there is very little in this paper that is new. Extensive profiling of RUNX1::RUNX1T1 AML for adults and children has been reported. If this paper is to be published the authors should make it very clear how their analysis makes an important contribution to the field.

A more minor concern is that it is not clear what is meant by “bottom 50” for DEGs. From the excel table it seems that these are genes that are not differentially expressed between fusion-positive and fusion-negative cases. Is the idea to find genes and programs that are similar between t(8;21) and other AMLs? Showing the GO functions etc that are enriched in non-DEGs is confusing. It is also not clear what the purpose of making a UMAP out of GO terms is.

Author Response

Reviewer 1

This is a data mining report. The authors accessed publicly available gene expression data for adults and children with t(8;21) AML. They identified DEGs between t(8;21)+ and negative independently for the two age groups, and then used various programs to infer regulatory networks, protein-protein interactions, etc. The datasets used are rather small. In the adult dataset there were only 7 patients with the fusion of interest, and only 16 in the pediatric data. Despite all the analysis reported, there is very little in this paper that is new. Extensive profiling of RUNX1::RUNX1T1 AML for adults and children has been reported. If this paper is to be published the authors should make it very clear how their analysis makes an important contribution to the field.

A more minor concern is that it is not clear what is meant by “bottom 50” for DEGs. From the excel table it seems that these are genes that are not differentially expressed between fusion-positive and fusion-negative cases. Is the idea to find genes and programs that are similar between t(8;21) and other AMLs? Showing the GO functions etc that are enriched in non-DEGs is confusing. It is also not clear what the purpose of making a UMAP out of GO terms is.

Author response: It is true that the two datasets that we used (GSE37642 and GSE75461) are rather small, with just 7 and 16 RUNX1-RUNX1T1-positive AML patients, respectively, out of a total of 140 adult and 48 pediatric AML patients. However, these are the only studies we could find in the GEO repository, providing gene expression data of such patients. It is true that our results recapitulate previous findings on the deregulated gene of RUNX1/RUNX1T1-positive AML patients, but they also provide new knowledge regarding differences and similarities between adult and pediatric RUNX1/RUNX1T1-positive AML patients, as well as regarding their gene regulatory networks.

By "bottom 50" DEGs, we mean the significantly downregulated genes in RUNX1/RUNX1T1-positive compared to RUNX1/RUNX1T1-negative AML patients, having the lowest levels of log2fold change (logFC), BH adjusted p-values and B values. In Table S1 we have included all the genes stemming from the analysis (19,527 in total), but by "bottom 50" we do not mean genes from rows #19,478 to #19,527, rather than from row #217 upwards (i.e., with adj. p-value< 5.0E-02). We believe that this was a misunderstanding and we hope to have clarified it.

In lines 75-83 of the Introduction, we make it very clear that our analysis makes an important contribution to the field, by acquiring a better understanding on the link between the DEGs and leukemic cells that harbor the RUNX1/RUNX1T1 fusion, as well as by deciphering differences between the two age groups of AML patients. We provide new information on the regulatory networks, composed of TFs and protein kinases, as well as the protein-protein interactions of the deregulated genes in RUNX1/RUNX1T1-positive adult and pediatric AML patients.

To gain a better understanding on the regulatory functions of the RUNX1/RUNX1T1 fusion oncogene in the transcriptome of AML patients of different age groups, we investigated the up-stream regulators of the differentially expressed genes (DEG) in two independent GEO datasets composed of adult and pediatric AML patients harboring the RUNX1/RUNX1T1 fusion oncogene. We then annotated and extracted gene expression signatures, to acquire a better insight of the link between the DEGs and leukemic cells that harbor the RUNX1/RUNX1T1 fusion, as well as to decipher any differences between the two age groups of patients. We emphasized on the deregulated TFs and the protein kinases that seem to act as “hubs” in the gene networks formed in RUNX1/RUNX1T1-positive AML cases.”

In addition, we make this clear in several points throughout the text, such as in:

Lines 207-208: “In order to get a better insight of the critical hubs that orchestrate the expression of the DEGs, we constructed the PPI networks.”

Lines 240-241: “Here, we explored the gene networks governing the top deregulated genes in adult and pediatric RUNX1/RUNX1T1 AML patients.”

Regarding the reviewer’s last comment, we did not aim to find genes and programs that are similar between t(8;21) and other types of AML. Moreover, the aim to construct UMAPs out of GO terms is to show how similar GO terms cluster together, providing a better visualization.

Reviewer 2 Report

The article “Deregulated gene expression profiles and regulatory networks in RUNX1/RUNX1T1-positive AML patients” by Kanellou and Zaravinos is a well-articulated paper, describing the impact of RUNX1/RUNX1T1 towards many genes in acute myeloid leukemia (AML) patients. The work is well written and organized, focusing on adult and pediatric patients. The results are clear, highlighting both upregulated and downregulated genes. For these reasons the article is suitable for publication after the following minor revisions:

- In the introduction, a proper section describing AML is missing. Author must describe the incidence of this disease, the main treatments and the constant research work in the field to emphasize the importance of the study. Recent article that should be cited is: Drug Development Research, 2022, 83(6), pp. 1331–1341. DOI 10.1002/ddr.21962

- Correct the numbering for Figure S1 and S2 in Figure 2 and Figure 4, respectively.

- The GO highlighted in figures 1 and 2 are not readable. Improve the quality of the image or format the text.

- Improve conclusions adding a brief summary of the most important genes up- and down-regulated.

- Remove Figure S1 and S2 from line 338 and 351, since already shown in the paper.

Author Response

Reviewer 2

The article “Deregulated gene expression profiles and regulatory networks in RUNX1/RUNX1T1-positive AML patients” by Kanellou and Zaravinos is a well-articulated paper, describing the impact of RUNX1/RUNX1T1 towards many genes in acute myeloid leukemia (AML) patients. The work is well written and organized, focusing on adult and pediatric patients. The results are clear, highlighting both upregulated and downregulated genes. For these reasons the article is suitable for publication after the following minor revisions:

- In the introduction, a proper section describing AML is missing. Author must describe the incidence of this disease, the main treatments and the constant research work in the field to emphasize the importance of the study. Recent article that should be cited is: Drug Development Research, 2022, 83(6), pp. 1331–1341. DOI 10.1002/ddr.21962

Author response: Would like to thank the reviewer for his/her positive feedback on our work. We have now included a section describing AML in the Introduction section, providing all the proposed information about incidence, treatment, etc.

Lines 41-61: “Acute myeloid leukemia (AML) is a heterogeneous disease in respect of molecular aberrations and prognosis. It is the second most common type of leukemia in adults and the most common type of acute leukemia. Nevertheless, AML is relatively rare and it accounts for approximately 1% of adult cancers in the United States, but nearly 2% of cancer-related deaths [1]. The Surveillance, Epidemiology, and End Results (SEER) Program reported annual incidence rates from 2010 and onward being consistently higher than 4.2 per 100,000 per year [2]. The overall AML incidence in childhood has also increased between 1975 and 2014 [3]. AML is defined by clonal expansion of malignant hematopoietic progenitor cells combined with differentiation arrest in the bone marrow [4]. Both epigenetic and genetic alterations have been implicated in the disease pathogenesis, with ~30-40% of the case to be associated with recurrent chromosomal translocations resulting in the generation of chimeric oncogenes [5].

In AML, drug resistance toward standard chemotherapeutic compounds is due to different factors, such as enhanced DNA repair, inhibition of cancer cell apoptosis, alteration of signaling pathways, alterations in drug metabolism enzymes, drug sequestration in intracellular organelles and overexpression of efflux drug transporters [6,7]. In addition, overexpression of P-gp, can also lead to failure of the treatment or relapse. P-gp overexpression can result in highly aggressive AML clones, which need more intensive treatment and invasive procedures [5,6]. Multidrug resistance mechanisms produce resistance to microtubule-targeting agents [7], such as Vinca alkaloids and Taxanes. Recently, three [1,2]oxazolo[5,4-e]isoindole derivatives showed antimitotic activity in HL-60R cells, by inhibiting tubulin polymerization, suggesting that they could represent a valuable tool to overcome MDR mechanism [8].

Lines 67-73: “The presence of t(8;21) establishes the diagnosis of AML, irrespective of blast count [10]. The clinical and prognostic features of t(8;21) differ between adults and children. In adults, the t(8;21) is found in 1-7% of adults with newly diagnosed AML and the age at presentation is younger than that for the overall group of adults with AML [11]. AML with the t(8;21) has a favorable prognosis in adults. In children, it is the most frequent chromosomal abnormality with AML. In contrast to adults, children with t(8;21) have poor outcomes, particularly when accompanied by additional cytogenetic abnormalities, such as del(9q), or gain of chromosome 4 [12].

- Correct the numbering for Figure S1 and S2 in Figure 2 and Figure 4, respectively.

Author response: We have revised the Figure numbering, as suggested by the reviewer. In detail, the Figure numbering reorganization is as follows: Figure S1 -> Figure 2; Figure S2 -> Figure 4; Figure 2 -> Figure 3; Figure 3 -> Figure 5; Figure 4 -> Figure 6.

- The GO highlighted in figures 1 and 2 are not readable. Improve the quality of the image or format the text.

Author response: We have now increased Figures 1 and 2 (as well as the rest of the Figures) to improve the visibility of the GO terms and rest information provided.

- Improve conclusions adding a brief summary of the most important genes up- and down-regulated.

Author response: We have now revised the Conclusion, as per the reviewer’s request.

“To conclude, our data corroborate that hematopoiesis is strictly regulated by different levels of inter-cellular signaling, and is affected by various intricate transcriptional regulations. The RUNX1-RUNX1T1 fusion aberrates many intracellular pathways, that involve tumor suppressors and oncogenes. The regulatory networks governing the top deregulated genes in adult and pediatric RUNX1/RUNX1T1 AML patients, include several transcription factors, intermediate proteins and kinases, which orchestrate the establishment and maintenance of leukemia. The most upregulated genes in adult RUNX1-RUNX1T1-positive AML involved RUNX1T1, POU4F1, CACNA2D2, FBLN5 and CAV1, which play a role in axonogenesis, ECM organization, and negative regulation of cell differentiation and ECM assembly. The top downregulated genes in this group included IGBP1, RNF39, FRMD1, INPPL1, FSCN2, NUBPL and UQCRQ, among others, playing a role in mitochondrial electron transport, ubiquinol to cytochrome c, aerobic electron transport chain and mitochondrial ATP synthesis coupled electron transport. The most upregulated genes in pediatric RUNX1-RUNX1T1-positive AML, included RUNX1T1, LINC00189, M1AP, ADCY7, TPPP3, ADARB1 and TSPAN32, playing key role in ECM organization, protein polymerization and cellular protein catabolic process. Finally, the top downregulated genes in this group, involved SHANK1, linc-RXFP2-1, SNORD9 and EGLN1, being primarily enriched in membrane depolarization during cardiac muscle cell action potential.”

- Remove Figure S1 and S2 from line 338 and 351, since already shown in the paper.

Author response: We thank the reviewer for pointing this. We have now removed the legends of Figures 1 and 2 from the Supplementary Materials, as suggested.

Reviewer 3 Report

The authors investigated the deregulated gene expression (DEG) profiles in RUNX1/RUNX1T1-positive AML patients and compared their functions and regulatory networks between adult and pediatric patients.

They demonstrated that the top-upregulated genes in (both adult and pediatric) RUNX1/RUNX1T1-positive AML patients are enriched in extracellular matrix organization, cell projection membrane, filopodium membrane and supramolecular fiber.

My comments are as follows:

1. Did your similarities arise in children and adults of all ages or in young adults/the elderly? Are you able to make this subgroup analysis? Are there any similarities between children and young adults? If so, would pediatric-inspired based treatment regimens benefit young adults with RUNX1-mutated AML?

2. How do epigenetic factors and redox-linked transcription factors impact the activation of RUNX1-dependent signaling pathways and consequently the evolution of children and adult patients with AML?  

3. A discussion regarding leukemic stem cells and their contribution to RUNX1-mutated AML is also warranted. See the following paper regarding AML LSCs: https://pubmed.ncbi.nlm.nih.gov/36163119/

Overall, well-written paper which can be accepted following minor revisions.

Author Response

Reviewer 3

The authors investigated the deregulated gene expression (DEG) profiles in RUNX1/RUNX1T1-positive AML patients and compared their functions and regulatory networks between adult and pediatric patients.

They demonstrated that the top-upregulated genes in (both adult and pediatric) RUNX1/RUNX1T1-positive AML patients are enriched in extracellular matrix organization, cell projection membrane, filopodium membrane and supramolecular fiber.

My comments are as follows:

  1. Did your similarities arise in children and adults of all ages or in young adults/the elderly? Are you able to make this subgroup analysis? Are there any similarities between children and young adults? If so, would pediatric-inspired based treatment regimens benefit young adults with RUNX1-mutated AML?

Author response: This is an interesting point. Our similarities arise between the two groups. i.e., pediatric and adult AML.

We have found that the average age of the adult RUNX1/RUNX1T1-positive AML patients (GSE37642 dataset) is 48.85 years, whereas for the RUNX1/RUNX1T1-negative ones, it is 55.95 years. Unfortunately, the exact age per pediatric patient is not provided in the GSE75461 dataset, and therefore, we cannot perform comparisons between young adults and children with RUNX1-mutated AML. If fact, the Table below shows the age groups in the GSE75461 study (Porcù E, et al. The long non-coding RNA CDK6-AS1 overexpression impacts on acute myeloid leukemia differentiation and mitochondrial dynamics. iScience. 2021;24(11):103350), with an average age of 7.4 years for the first three quartiles and 10.1 years for the forth quartile:

Clinical and biological features of AML patients

CDK6-AS1 RQ-exp

I+II+III quartile

IV quartile

#p-value

TOTAL n=132

100

32

Gender (Female/Male)

ns

F

38 (38%)

11 (34%)

M

62 (62%)

21 (66%)

Age at diagnosis

0.006

<2 years

27 (27%)

1 (3%)

2-10 years

36 (36%)

11 (34.5%)

>10 years

37 (37%)

20 (62.5%)

Mean

7.4

10.1

  1. How do epigenetic factors and redox-linked transcription factors impact the activation of RUNX1-dependent signaling pathways and consequently the evolution of children and adult patients with AML?  

Author response: In the following parts we now discuss different epigenetic factors and redox-linked transcription factors that impact the activation of RUNX1-dependent signaling pathways and consequently the evolution of children and adult patients with AML, as suggested by the reviewer.

Lines 238-240: “Leukemia is characterized by somatic mutations and genetic rearrangements that impede signal transduction and expression programs, resulting in the disruption of chromatin modifiers and transcription factors. In AML, transcriptional and epigenetic reprogramming can be seen in the RUNX1 transcription factor and its oncogenic derivative RUNX1/RUNX1T1 [34].

Lines 328-354: “A large number of studies have recently revealed that AML patients have repeated mutations in several important epigenetic mediators [81]. Mutations in genes involved in DNA methylation, such as DNA methyltransferase 3A (DNMT3A) and isocitrate dehydrogenase 1 and 2 (IDH1/2), are the most common. The majority of these mutations affect the hematopoietic stem cells (HSC) compartment and alter hematopoietic differentiation, in some cases leading to myelodysplasia, stem cell expansion, or other preleukemic conditions [82–84]. In addition, mutations in other epigenetic modifiers, such as additional sex combs-like (ASXL1) and enhancer of zeste homolog 2 (EZH2), contribute to the occurrence of AML. Nevertheless, epigenetic mutations alone are insufficient to transform HSCs, implying that mutations must be acquired sequentially [25]. In Npm1c/Dnmt3a mutant knock-in mice, a model of AML development, the use of a small molecule (VTP-50469) could indeed, reverse myeloid progenitor cell self-renewal prior to leukemia transformation, suggesting that individuals at high risk of developing AML may benefit from preventive targeted epigenetic therapy [85].

The effect of RUNX1 on target gene expression is highly context-dependent, determined by the composition of the transcriptional complexes in which RUNX1 functions at a particular gene. These complexes contain transcriptional cofactors as well as epigenetic modifiers, which are recruited to gene promoters as well as other regulatory regions, and by doing so, RUNX1 can influence both transcriptional activity as well as the epigenetic status of target genes [34].

RUNX1 has been found to complex with an array of epigenetic modifiers, with the outcome for both RUNX1 function and target gene activity dependent on the balance of activating and repressive factors associated with RUNX1 at a particular time. These studies thus build a picture of RUNX1 it-self as a target of epigenetic modifiers, which post-translationally modify the RUNX1 protein, altering its coactivator/corepressor interactions and thus influencing its transcriptional activity. In addition, RUNX1 recruits these coactivators and corepressors to target genes, resulting in modification of the chromatin environment and thus affecting gene activity at this second level. An elegant example of this multi-layered interaction between RUNX1, epigenetic modifiers and the epigenome is demonstrated in the recently described interaction of RUNX1 with protein arginine methyltransferase 6 (PRMT6) [34].

  1. A discussion regarding leukemic stem cells and their contribution to RUNX1-mutated AML is also warranted. See the following paper regarding AML LSCs: https://pubmed.ncbi.nlm.nih.gov/36163119/

Author response: We thank the reviewer for suggesting to include this. We have now inserted the following paragraph in lines 321-332, and cited Chen Y et al. Cell Death Discov. 2022.26;8(1):397. doi: 10.1038/s41420-022-01193-0, as suggested.

A large number of studies have recently revealed that AML patients have repeated mutations in several important epigenetic mediators [79]. Mutations in genes involved in DNA methylation, such as DNA methyltransferase 3A (DNMT3A) and isocitrate dehydrogenase 1 and 2 (IDH1/2), are the most common. The majority of these mutations affect the hematopoietic stem cells (HSC) compartment and alter hematopoietic differentiation, in some cases leading to myelodysplasia, stem cell expansion, or other preleukemic conditions [80–82]. In addition, mutations in other epigenetic modifiers, such as additional sex combs-like (ASXL1) and enhancer of zeste homolog 2 (EZH2), contribute to the occurrence of AML. Nevertheless, epigenetic mutations alone are insufficient to transform HSCs, implying that mutations must be acquired sequentially [25]. In Npm1c/Dnmt3a mutant knock-in mice, a model of AML development, the use of a small molecule (VTP-50469) could indeed, reverse myeloid progenitor cell self-renewal prior to leukemia transformation, suggesting that individuals at high risk of developing AML may benefit from preventive targeted epigenetic therapy [83].

Additionally, in lines 358-360 we have now inserted the following sentence: “EZH2 controls the expression of genes involved in stem cell maintenance and differentiation and its down-regulation inhibits apoptosis, affects the expression of MAD2 and CDC20, and promotes chromosomal instability in AML cells”.

Overall, well-written paper which can be accepted following minor revisions.

Author response: We would like to thank the reviewer for his/her constructive comments and overall favorable opinion in our work.

Round 2

Reviewer 1 Report

Thank you for clarifying that the "bottom 50" DEGs are the 50 most differentially expressed genes that are decreased in the fusion-positive cases compared to controls.

Regarding the use of two small datasets for this project, it is not clear why the authors could not access other public data. There are several quite large public repositories of AML gene expression data, such as TCGA for adults and TARGET for pediatrics, to name only two.

Regarding the claim that this paper provides new insight into differences between adult and pediatric cases, the authors do not directly compare adult v. pediatric cases to determine differences, but rather merely present the analyses for the two age groups side by side.

Reviewer 2 Report

The authors properly reviewed the manuscript, hence it is suitable for publication.